# Beyond Parenting Socialization Years: The Relationship between Parenting Dimensions and Grandparenting Functioning

**DOI:** 10.3390/ijerph19084528

**Published:** 2022-04-09

**Authors:** Sofia Gimenez-Serrano, Marta Alcaide, Maria Reyes, Juan J. Zacarés, Montserrat Celdrán

**Affiliations:** 1Department of Developmental and Educational Psychology, Faculty of Psychology, University of Valencia, 46010 Valencia, Spain; sogise@alumni.uv.es (S.G.-S.); maria.f.reyes@uv.es (M.R.); 2Department of Methodology of Behavioral Sciences, Faculty of Psychology, University of Valencia, 46010 Valencia, Spain; maralna2@alumni.uv.es; 3Department of Cognition, Development and Educational Psychology, Faculty of Psychology, University of Barcelona, 08035 Barcelona, Spain; mceldran@ub.edu

**Keywords:** parenting, grandparenting, satisfaction with life, older adults, grandchildren

## Abstract

Parental socialization has been studied mainly when is in process, but less is known about its long-term impact on older adults, particularly on one of the most important developmental tasks in later life: being a grandparent. Participants were 313 Spanish grandparents. The present study examined the relationship between parenting and its impacts in the long term, when the child is a grandparent. The variables examined were parenting (parental warmth and parental strictness) and grandparenting functioning (satisfaction with life, meaning of life, parent–adult child relationship quality, emotional closeness with grandchildren, and role overload). The statistical analyses were a correlation analysis and multiple linear regression analyses. A constant pattern between parenting and grandparenting functioning has been found. Warmth was positively associated with grandparenting functioning, as opposed to strictness, which did not show benefits for grandparents and even showed a significant negative relationship with an indicator of grandparenting functioning. Present findings highlight that, during the socialization years, greater parental warmth but not parental strictness might be of benefit for children at the end of their life (i.e., when they are grandparents) but also for their descendants because they have a better relationship with the two following generations (i.e., adult children and grandchildren).

## 1. Introduction

### 1.1. Parental Socialization

In the study of parental socialization, the practices of parents are often classified around two main dimensions: warmth (e.g., dialogue, support, or displays of affection) and strictness (e.g., practices like demand, setting limits, punishments, or discipline), based on a two-dimensional model [1,2]. Warmth represents parental love, approval, acceptance, and support [3,4,5], whereas strictness refers to the discipline from the parents towards their children, controlling and/or supervising their behavior [6,7,8]. The two-dimensional model is based on theoretically orthogonal dimensions (i.e., non-related) that gives origin to four parenting styles: authoritarian (strictness but not warmth); indulgent (warmth but not strictness); authoritative (strictness and warmth); neglectful (neither strictness nor warmth) [1,2].

Two main parenting dimensions (i.e., warmth and strictness) have been studied in the scientific literature to identify their impact on child development [9]. Traditionally, the parenting strategy based on warmth and strictness (i.e., so-called authoritative parenting) has been associated with optimal child development for the different criteria. For example, some studies reported that parental strictness combined with parental warmth is related to higher levels of emotional maturity, psychosocial competence, academic performance [1,10,11,12,13], secure attachment [14], as well as a lower incidence of emotional disturbances and behavioral problems [9]. However, much of this research is focused on Anglo-Saxon contexts with mainly European-American samples. When analyzing the cultural variability to explain the optimal parenting, the impact of parenting is not always the same in all cultural contexts. For example, in a systematic review on the subject, parenting based on strictness without the warmth dimension (i.e., so-called authoritarian parenting) is associated with greater outcomes in some domains of adjustment among ethnic minorities from the United States, namely Chinese-Americans and African-Americans, as well as in Arab societies or some Asian countries [9]. Overall, culture plays an important role in the psychological process [15,16]. In particular, these discrepancies about the optimal parenting strategy may point to the importance of cultural factors on the impact of parental socialization [17,18,19]. Parenting defined by warmth combined with strictness may not always benefit child development in all cultural contexts [9,20,21].

Additionally, findings from emergent research in European and Latin American contexts have also pointed out that parenting based on warmth but not strictness (i.e., so-called indulgent parenting) provides equal or even greater benefits than parenting based on warmth and strictness (i.e., authoritative parenting). In these studies, children with parents who are warm but not strict (i.e., indulgent) have higher scores on self-concept [22,23,24], psychosocial maturity [25], benevolent social values [23], and connectedness with nature [24]. They also show lower levels of aggressiveness, emotional instability, behavioral problems [21,25,26], and less hostile sexism [23,27].

The impact of parental socialization affects children, and adolescents [21,22] but also adult children [23,26], despite the fact that parental socialization is over when the adolescent child reaches the adult age. Nevertheless, most of the scientific literature has focused its studies on the first stages of the life cycle, i.e., when parental socialization is in progress and parents are raising their children [28,29], based on the premise that childhood and adolescence are the stages of greatest plasticity [25,30]. However, the process underlying parental socialization could also be quite relevant for adult children over the years, despite the different and multiple influences across the life cycle, particularly in adulthood and later life. In fact, parenting might have an impact beyond adolescence and differences in adjustment among adult children could be also related to parenting patterns of warmth and strictness. For this reason, a growing body of researchers are trying to study the impact of parental socialization on individuals throughout the life cycle.

### 1.2. Parental Socialization at the End of the Life Cycle

The literature specializing in parenting and the consequences for adjustment in the later stages of the life cycle is scarce in comparison with the other stages. There are only a few studies that combine samples from different ages. For example, in a study with non-consecutive stages of the life cycle, with adolescents (aged 12–17 years) and adult children (aged 60–75 years), the adolescents and older adults with warm parents (i.e., indulgent, and authoritative) reported higher self-esteem and values of self-transcendence and conservation than their counterparts from families who were not based on warmth (i.e., authoritarian and neglectful). However, the highest levels of social, emotional, and family self-esteem were found in the warm but not strict family group (i.e., indulgent parenting), regardless of age [31]. In another similar study, covering all stages of the life cycle, the analysis found that higher parental warmth and lower parental strictness (i.e., indulgent parenting) in the group of older people was associated with higher scores in family self-esteem, self-competence, social competence, and empathy, and in other age groups the pattern was also quite similar [25]. These findings suggest that, as in adolescence, older adults with greater adjustment might be those who were raised by families characterized by warmth, whereas parental strictness could be unnecessary or even harmful for adjustment in later life. Similar results were obtained in another study with the same age groups [26]. The group of older people with the lowest rates of aggression corresponded to those from warm families (i.e., authoritative and indulgent), but only older people raised by warm but not strict families (i.e., indulgent) had the highest scores on family self-concept and on social adjustment criteria such as the internalization of universal values. Another study, which also included older adults, found benefits for adjustment related to parental warmth, whereas parental strictness was found to be unnecessary or even harmful [23]. Interestingly, higher parental warmth and lower parental strictness (i.e., indulgent parenting) significantly influenced several key social adjustment criteria in older people. Specifically, it was associated with higher levels of internalization of human values as well as lower rates of hostile sexism.

The abovementioned studies agree on the link between parenting and social and psychological adjustment criteria in the short term (when parental socialization is in progress) and in the long term (when parental socialization is over). In addition, these studies seem to suggest that warmth but not strictness (i.e., indulgent parenting) was associated with equal or even better socialization outcomes than warmth and strictness (i.e., authoritative parenting), while a lack of warmth (i.e., authoritarian and neglectful parenting) was associated with the worst socialization outcomes in all areas [23,25,26,31].

There is little evidence regarding parental socialization focused on the last stage of the life cycle. Additionally, within the few studies examining parenting and adjustment among adult children in later life, the adjustment was captured without considering one of the main developmental tasks of later life: being a grandparent. Interestingly, grandparenting represents a family intergenerational continuity: grandparents (G1), adult children (G2) and grandchildren (G3). Overall, grandparents are a main source of care, protection, and significance to the grandchild, even though the primary caregivers are usually the parents rather than the grandparents [32,33,34]. Unfortunately, not all grandparents develop a good relationship with their adult children as well as good grandparenting functioning based on an emotional closeness to the grandchild, but low role overload. Differences in competence and adjustment as well as in grandparenting functioning among older adults could be related to the first experiences of the older adult in their family.

### 1.3. Present Study

The main objective of this study was to analyze the relationship between parenting dimensions (i.e., warmth and strictness) with different indicators of grandparenting functioning: (i) satisfaction with life, (ii) meaning of life, (iii) parent–adult child relationship quality, (iv) emotional closeness with grandchildren, and (v) role overload. In the present study, it is hypothesized that parental warmth would be associated with better scores in grandparenting functioning, i.e., higher scores on satisfaction with life, meaning of life, parent–adult child relationship quality, and emotional closeness with grandchildren; and lower scores on role overload. It also was expected that parental strictness would not be related to grandparenting functioning.

## 2. Materials and Methods

### 2.1. Participants and Procedure

The study was composed of 313 grandparents (243 females and 70 males; *M* = 70.73 years, *SD* = 5.59) aged from 60 to 89 years old, who have a grandchild aged from 1 to 19 years old (*M* = 8.68 years, *SD* = 4.38) and take care of them, but not as a caregiver. It was carried out in three large cities of Spain (Barcelona, Seville, and Valencia). The participants were recruited from community senior centers, as in other previous studies [25,31]. A random selection of senior centers was performed from the complete list of centers. Directors of senior centers were invited to participate in the research. If an institution refused to participate, a replacement institution was randomly selected until the required sample size was obtained. This random sampling procedure means that the probability of each unit in the population being selected is the same [35,36,37]. The distribution by educational level was as follows: no studies: 5.1% (*n* = 16); primary studies: 24.6% (*n* = 77); secondary studies: 30.7% (*n* = 96); university studies: 39.6% (*n* = 124).

As in previous family studies with older adults, the grandparents were the respondents [38,39,40]. The research protocol was approved by the research ethics committee of the University of Barcelona (Institutional Review Board, IRB00003099). For all participants, their participation was voluntary, informed consent was required, and anonymity of responses was guaranteed.

### 2.2. Measures

#### 2.2.1. Parenting

Parenting was captured in terms of the two main dimensions: warmth and strictness. Warmth was measured with the adult version of the Warmth/Affection Scale (WAS) [41]. This scale measures the extent to which the older adults (in this case, grandparents) perceived their parents as loving, responsive, and involved; for example, “Let me know they loved me”. Responses were on a 4-point scale from 1 (“almost never true”) to 4 (“almost always true”). The alpha value was 0.957. Strictness was measured with the adult version of the Parental Control Scale (PCS) [41]. The PCS measures the extent to which the older adults (in this case, grandparents) perceived strict parental control over their behavior; for example, “Were always telling me how I should behave”. The alpha value was 0.873. Both WAS and PCS scales are reliable and valid measures for adult children to assess parental socialization, when they were raised by their parents during the socialization years. The two scales (WAS and PCS) are widely used in the literature to measure parental socialization in adult children [31,42,43,44]. Higher scores on the WAS and PCS scales represent a greater sense of parental warmth and parental strictness [45,46].

#### 2.2.2. Grandparent Adjustment

Satisfaction with life was measured with five items from the Satisfaction with Life Scale (SWLS) [47,48]. This scale is focused on assessing global life satisfaction, an indicator of hedonic well-being; for example, “In most ways my life is close to my ideal”. Items were answered in a 7-point scale from 1 (“strongly disagree”) to 7 (“strongly agree”). The SWLS has favorable psychometric properties [47], including high internal consistency and high temporal reliability. It correlates moderately to highly with other measures of subjective well-being. This scale is suited for use with different age groups, including older adults [43,47]. Higher scores are related to greater satisfaction with life. The alpha value was 0.808.

The meaning of life was measured with the reduced Spanish version with 10 items [49] from the Purpose in Life Test (PIL Test) [50,51]. This measure assesses the meaning of life and it has good psychometric properties characterized by an acceptable degree of reliability and validity [52]; for example, “Life to me seems:”, with options ranging from “always exciting” to “completely routine”. To answer each item, the person places him/herself on a scale from 1 to 7 between two extreme feelings: 1 being related to a poor meaning of life and 7 related to a greater meaning of life. The reduced Spanish version of the PIL Test [49] has been used in studies with older adults [53]. The authors of the instrument proposed the meaning of life as a global concept and it was subsequently corroborated in different studies using confirmatory factor analyses [54,55]. The meaning of life has been related with eudaimonic well-being, positive affect, adequate coping, and happiness [56,57,58,59,60]. High scores on this measure represent a greater meaning of life. The alpha value was 0.905.

#### 2.2.3. Grandparents and Their Adult Children

Parent–adult child relationship quality was measured with the Family APGAR questionnaire [61]. This is a scale aimed at assessing family relationships, but it was specifically adapted to evaluate the parent-child relationship [62]; for example, “I am satisfied that I can turn to my children for help when something is troubling me”. The instrument allows three possible responses (2, 1, 0) to each of the five items in the questionnaire. Higher scores were related to a high satisfaction with family function. The alpha value was 0.695.

#### 2.2.4. Grandparents and Their Grandchildren

Emotional closeness with grandchildren and role overload were measured with two subscales (i.e., rewards and stressors) of the Spanish version [63] of the Parental Stress Scale (PSS) [64], adapted for grandparents. It is common in the literature to assess the parental stress [65], but in this case an adaptation was made to assess the grandparenting role [66]. Previous studies have used the PSS [67] and other similar questionnaires [40,68], adapted for grandparents instead of parents. In this case, an adaptation was made to the PSS, in which respondents, instead of being “parents”, had been “grandparents”, such as in other previous research [67]. Items were answered on a 4 Likert-type response scale ranging from 1 (strongly disagree) to 4 (strongly agree). Emotional closeness with grandchildren was measured with the Rewards subscale of the PSS [63]. This subscale assesses satisfaction as a grandparent and closeness towards the grandchild; for example, “I feel happy in my role as a grandparent”. Higher scores indicated a higher level of emotional closeness with their grandchildren. The alpha value was 0.746. Role overload was measured with the Stressors subscale of the PSS [63]. This subscale measures discomfort and difficulties in the role of grandparent; for example, “Taking care of my grandchild sometimes takes more time and energy than I have”. Higher scores indicated a higher level of role overload. The alpha value was 0.846.

### 2.3. Data Analysis

A correlation analysis and multiple linear regression analyses were performed. The correlation analysis was conducted between the two main dimensions of parenting (i.e., warmth and strictness) and grandparenting functioning (i.e., satisfaction with life, meaning of life, parent–adult child relationship quality, emotional closeness with grandchildren, and role overload). Additionally, a correlation between the different indicators of grandparenting functioning was applied. Furthermore, a lineal regression was applied in which the dependent variables were those related to grandparenting functioning (i.e., satisfaction with life, meaning of life, parent–adult child relationship quality, emotional closeness with grandchildren, and role overload) and the predictors were the two main dimensions of parenting (i.e., warmth and strictness).

## 3. Results

Results from correlation analysis between parenting dimensions and grandparenting functioning are presented in Table 1. Some statistically significant correlations were found between parenting dimensions and grandparenting functioning. Parental warmth was positively associated with grandparenting functioning. Grandparents who scored higher in terms of parental warmth had greater levels of satisfaction with life, meaning of life, parent–adult child relationship quality, and emotional closeness with grandchildren. In contrast, parental strictness did not show a correlation with grandparenting functioning and was even harmful. Greater scores in parental strictness were related to a greater role overload. Additionally, some grandparenting functioning dimensions were positively correlated between them. Specifically, higher scores in satisfaction with life were associated with a greater meaning of life. In turn, satisfaction with life and meaning of life had a positive relationship with the parent–adult child relationship quality and emotional closeness with grandchildren (these last variables were related to each other).

Results from linear multiple regression analysis were similar to those obtained in the correlation analysis. The results for the predictions in grandparenting functioning (i.e., satisfaction with life, meaning of life, parent–adult child relationship quality, emotional closeness with grandchildren, and role overload), depending on parenting dimensions (i.e., parental warmth and parental strictness) are presented in Table 2. A multiple linear regression model for each indicator of grandparenting functioning was performed. Overall, parenting tended to predict grandparenting functioning. Interestingly, a common pattern was found for the five predictions models.

Parental warmth was a significant predictor of grandparenting functioning whereas parental strictness did not reach the statistically significant level. For the prediction of satisfaction with life, only parental warmth was a significant predictor, whereas parental strictness did not reach the statistically significant level. Greater parental warmth positively predicts satisfaction with life in grandparents. For the prediction of meaning of life, parental warmth and parental strictness were not significant predictors. For the prediction of parent–adult child relationship quality, a similar association as for satisfaction with life was found. Again, parental warmth was a significant predictor, but not parental strictness. Scores in the parent–adult child relationship quality were positively predicted by parental warmth. Similarly, for the prediction of emotional closeness with grandchildren, parental warmth was a significant positive predictor, but not parental strictness. Finally, for the prediction of role overload, any of the two parenting dimensions (i.e., parental warmth and parental strictness) reached the significant statistical level. However, a higher explained variance was found in terms of satisfaction with life, in comparison to the other indicators of grandparenting functioning (i.e., meaning of life, parent–adult child relationship quality, emotional closeness with grandchildren, and role overload). Satisfaction with life was the most predicted model by parental warmth.

## 4. Discussion

In this study, we analyzed the influence of the two main dimensions of parenting (i.e., warmth and strictness) of a two-dimensional model [2] on grandparenting functioning, captured by measuring the satisfaction with life, meaning of life, parent–adult child relationship quality, emotional closeness with grandchildren, and role overload. Correlation and multiple linear regression analyses have shown significant findings. A constant pattern between parenting and grandparenting functioning has been found. Warmth was positively associated with grandparenting functioning, as opposed to strictness, which did not show benefits for grandparents and even showed a significant negative relationship with an indicator of grandparenting functioning.

Some interesting findings have been obtained in this study. There were positive relationships between different dimensions of grandparenting functioning. Satisfaction with life and meaning of life were identified in previous studies as two important indicators of health and well-being in later life [53,69]. However, less is known about the correlation between satisfaction with life and meaning of life in a specific group of older adults: those who are grandparents. Interestingly, according to the present findings, satisfaction with life and meaning of life in grandparents are connected to healthy relationships with the following generations: the second generation (i.e., adult children) but also the third generation (grandchildren).

Specifically, according to the findings, those grandparents with greater well-being (i.e., higher satisfaction with life and meaning of life) reported a good relationship with the second generation (i.e., parent–adult child relationship quality), and the third generation (i.e., greater emotional closeness with grandchildren). It should be noted that these findings from the present study do not agree with some theorists, who suggest that grandparents with greater well-being might have a positive relationship with their grandchildren regardless of the relationship with their adult children (or even with a negative relationship with their adult children). In contrast, the present findings seem to support the idea that grandparents with greater well-being (e.g., satisfaction with life and meaning of life) are not only those who may view grandchildren, but also adult children, as a continuation of their lives and experience greater affection (see [70,71]).

Another central result of this study is the different influence of the main parenting dimensions (i.e., warmth and strictness) of the two-dimensional model [1,2] on grandparenting functioning. Interestingly, differences between grandparents (i.e., in satisfaction with life and meaning of life), as well as the relationship with their adult children (i.e., parent–adult child relationship quality) and grandchildren (i.e., emotional closeness with grandchildren and role overload) were consistently related to the family in which the older adults were raised. On the one hand, parental warmth has been associated with a higher degree of satisfaction with life, meaning of life, parent–adult child relationship quality, and emotional closeness with grandchildren, according to correlational analysis. The same tendency was also found in the predictions of grandparenting functioning with parental warmth as a predictor, except for meaning of life (which did not reach the statistically significant level). On the other hand, parental strictness did not contribute to grandparenting functioning and even had some detrimental consequences. A positive correlation has been found between parental strictness and role overload, which may indicate that those grandparents who were raised by families that exercised control and surveillance over them during the socialization years, now seem to experience great overload in their role as grandparents.

Therefore, according to the findings from the present study, parental socialization could indirectly have a crucial impact beyond one generation. Remarkably, during the socialization years, greater parental warmth but not parental strictness might be beneficial for children at the end of their life (i.e., when they are grandparents), and also for their descendants because they have greater relationship with the two following generations (i.e., adult children and grandchildren). A few previous studies have also analyzed the parental socialization and its impact on older adult adjustment [26,31]. The present study also examines the consequences of parenting for older adults (satisfaction with life and meaning of life) but also goes beyond this, because they are focused on being a grandparent, one of the most important developmental tasks for older adults. Even if the grandparent’s socialization took place a long time ago, these effects last and affect both the present (grandparents) and indirectly the following generations (adult children and grandchildren).

Overall, the present findings suggest that parenting based on warmth during the socialization years offers broad benefits for grandparenting functioning, while parenting strictness seems to be unnecessary or even harmful. The results of this study do not completely coincide with classical studies based on the two-dimensional model [2] conducted on predominantly European-American middle-class families from the United States. Specifically, parental strictness (accompanied by warmth) was associated with greater benefits for adjustment and competence (e.g., a higher level of emotional maturity, psychosocial competence, and secure attachment). In contrast, the findings from the present study agree with the other studies based on the two-dimensional model [2] conducted mainly in European and Latin American countries. Specifically, parental warmth has been associated with good socialization outcomes, while parental strictness was identified as unnecessary or even harmful for child competence. However, these studies mostly included adolescents [72,73,74] and less studies are focused on older adults [26,31]. Therefore, the present study offers new evidence extending the benefits of parental warmth to grandparenting functioning in those older adults who have reached one of the most important developmental tasks in later life: being a good grandparent.

The present study has some important strong points, but some limitations should be considered. First, this study examined family socialization using a consolidated theoretical framework: the two-dimensional model [1,2]. Parental warmth during the socialization years is always beneficial for grandparenting functioning, whereas parental strictness seems to be unnecessary or even harmful. Additionally, this study adds new evidence focused on older adults, and, specifically to grandparenting functioning based on five indicators: grandparent adjustment (i.e., satisfaction with life and meaning of life), as well as a good relationship with the second generation (i.e., parent–adult child relationship quality), and the third generation (i.e., emotional closeness with grandchildren and role overload). Grandparenting functioning is influenced by a multitude of psychological and environmental factors that will need further analysis. For example, all participants came from community senior centers, in which a positive impact on health and social aspects has been seen previously [75,76]. However, within this group, the study still found differences relating to grandparenting functioning and the impact on participants’ well-being, depending on parenting during the socialization years. Another limitation of this study was related to the high degree of feminization of the sample. This is closely related to the characteristics of the participants that attend community senior centers, in which older women tend to participate more than men [77].

Focusing attention on older people has allowed the exploration of the consequences of parental socialization beyond adolescence [26,31], as well as its influence on relationships with other generations (indirectly, based on the relationship between the grandparents with their adult children and grandchildren). Taking into account the possible long-term impact of parental socialization on grandparents could help family counsels, in order to facilitate systemic changes to improve intergenerational relationships within three-generation families [78]. Moreover, geropsychologists could use narrative therapy in order to help grandparents who have had malfunctioned parental roles during their childhood to understand this connection and how it relates to their children and grandchildren [79].

Due to the long duration of time between parental socialization (when the older adult was raised by their parents) and the present time of being a grandparent (an adult child in the later stage of their life span), caution is advised when inferring cause–effect, as the study is not based on longitudinal data. However, a consistent pattern between parenting dimensions (i.e., warmth and strictness) and grandparenting functioning was found. Some studies have considered the differences in parenting practices between generations [43]. Nevertheless, less is known about the fact that parental socialization affects both the individual and the other generations (indirectly). Additionally, statistical results from regression analyses are quite similar to those from correlation analysis (although two significant differences obtained in correlation analysis were not found in regression analysis). In addition to the fact that future studies on grandparenting should use measures and adopt designs to overcome these limitations, other suggestions for future research can be noted. Future studies should examine parenting and its consequences with a greater sample [80]. However, a similar pattern was found according to both statistical analyses: parental warmth was positively associated with grandparenting functioning whereas parental strictness was unnecessary or even detrimental to grandparenting functioning. Likewise, it would be necessary to deepen the understanding of psychological processes through which parental socialization continues to influence long-term family relationships. For this, analysis of the internal working models of attachment of older adults should be introduced in order to show how parenting dimensions are associated with attachment styles in their current family interactions. Finally, measures of the continuity of parental socialization should be obtained through parenting styles and practices as grandparents.

## 5. Conclusions

Based on the two-dimensional model of parental socialization [2], the present study has found a consistent relationship between parental warmth and strictness during socialization years and grandparenting functioning, captured by satisfaction with life, meaning of life, parent–adult child relationship quality, emotional closeness with grandchildren and role overload. Overall, present findings highlight the beneficial role of the parental warmth dimension associated with better grandparenting functioning, but parental strictness dimension appears as unnecessary or even detrimental grandparenting functioning. Results obtained in the present study coincide with evidence from European and Spanish-speaking contexts [20,21,73]. Nevertheless, recent findings do not coincide with other studies conducted in Anglo-Saxon contexts in which only children with parents who are strict, and warm (i.e., authoritative parenting) have greater adjustment [1,11,13]. In addition, this study has examined some variables characteristic of older adults, which have not been widely used in the scientific literature [26]. This point has helped to understand the internal processes of grandparenting functioning, including well-being of grandparents (e.g., satisfaction with life and meaning of life) as well as family relationships with adult children and grandchildren. The factors that affect both hedonic and eudaimonic well-being in old age are of a very varied nature, both personal and relational and associated with the context [81,82]. In this study, only parental socialization has been considered in order to identify its possible influence on current well-being. Despite the amount of influences across life cycle [44], there was found a relation between grandparenting functioning and the family in which grandparents were raised. Grandparents who were raised by parents characterized by warmth have greater grandparenting functioning than those who were raised by parents without warmth. The contributions of the present study are relevant for the literature about parental socialization’s impact throughout the life cycle, increasing the evidence of a common pattern in the long term and not only in the short term [26,31] and focusing on the impact on both the individual and family relationships between different generations.

## Figures and Tables

**Table 1 ijerph-19-04528-t001:** Correlations between parenting dimensions and grandparenting functioning.

	1	2	3	4	5	6	7
1. Parental warmth	1						
2. Parental strictness	−0.097	1					
3. Satisfaction with life	0.216 **	−0.094	1				
4. Meaning of life	0.121 *	0.012	0.439 **	1			
5. Parent–adult child relationship quality	0.194 **	0.020	0.259 **	0.231 **	1		
6. Emotional closeness with grandchildren	0.132 *	0.101	0.270 **	0.222 **	0.245 **	1	
7. Role overload	0.027	0.138 *	−0.066	−0.103	0.012	−0.025	1

* *p* < 0.05; ** *p* < 0.01.

**Table 2 ijerph-19-04528-t002:** Multiple linear regression coefficients between parenting dimensions and grandparenting functioning.

Dependent Variable	Predictors	*B*	SE B	β	*t*	Lower	Upper
Satisfaction with life *R*^2^*_adj_*= 0.048 *F*(2, 287) = 8.22 ***							
	Parental warmth	0.06	0.02	0.21	3.63 ***	0.029	0.099
	Parental strictness	−0.06	0.04	−0.09	−1.48	−0.139	0.020
Meaning of life *R*^2^*_adj_* = 0.007 *F*(2, 287) = 1.96							
	Parental warmth	0.12	0.06	0.12	1.97	0.000	0.246
	Parental strictness	0.05	0.14	0.02	0.32	−0.234	0.325
Parent–adult child relationship quality *R*^2^*_adj_* = 0.032 *F*(2, 287) = 5.72 ***							
	Parental warmth	0.04	0.01	0.20	3.37 ***	0.017	0.064
	Parental strictness	0.02	0.03	0.03	0.55	−0.039	0.069
Emotional closeness with grandchildren *R*^2^*_adj_* = 0.017 *F*(2, 287) = 3.53 *							
	Parental warmth	0.02	0.01	0.12	2.00 *	0.000	0.045
	Parental strictness	0.05	0.03	0.11	1.92	−0.001	0.101
Role overload *R*^2^*_adj_* = 0.010 *F*(2, 287) = 2.51							
	Parental warmth	0.02	0.03	0.03	0.59	−0.042	0.078
	Parental strictness	0.15	0.07	0.13	2.21	0.017	0.292

* *p* < 0.05; *** *p* < 0.001.

## Data Availability

All data are available in this manuscript.

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
