# Peer review of "Beyond Parenting Socialization Years: The Relationship between Parenting Dimensions and Grandparenting Functioning"

_ijerph, 2022, doi:10.3390/ijerph19084528_

Round 1
Reviewer 1 Report
Thank you very much for the opportunity to review this interesting text. The article is well written, all theses are clearly presented, and the models used are quite adequate to the assumptions of the study.
The main objective of the presented in the paper study was to analyze the relationship between parenting dimensions (i.e., warmth and strictness) with different indicators of grandparenting functioning: satisfaction with life, meaning of life, parent-adult child relationship quality, emotional closeness with grandchildren, and role overload. As Authors state “According to findings from present study, parental socialization could indirectly have a crucial impact beyond one generation. Remarkably, during socialization years greater parental warmth, but not parental strictness, might be benefit for children at the end of their life (i.e., when they are grandparents) but also for their descendants because they have greater relationship with the two following generations (i.e., adult children and grandchildren). A few previous studies also have analyzed the parental socialization and its impact on older adult adjustment.” As true as this statement is, I would like to point out that it requires deeper explanation- what psychological mechanism is responsible for such results.
“This input has helped to understand the internal processes of grandparenting functioning, including well-being of grandparents (e.g., satisfaction with life and meaning of life) as well as family relationships with adult children and grandchildren.”
Satisfaction with life and meaning of life are influenced by many factors which should be addressed in the limitation of the study (this section is missing in the paper).
I would like to highlight a few more elements that can be improved:
- Explanation of how the gender disproportion of the respondents influenced the results. This issue should also be addressed in the limitations of the research.
- Due to the age of the respondents, it might be sensible to introduce some measure of cognitive functioning. which of course is not possible to do after the end of the study, but is a hint for the future research, and the limitation of the presented study.
- The analysis of the relationship between dimensions that concern the quality of life of grandparents requires more in-depth analysis. This relationship seems obvious, but it would certainly be good to try to explain the psychological mechanisms that explain it.
- I would like to emphasize that general measures, i.e. scales measuring mental well-being and meaning in life, are not directly related to being a grandparent. This issue should be included in the discussion of the results. It also shows the lack of data in the text on other characteristics than the age and sex and the activity in senior centers. Nothing is known about the structure of education among the respondents, and the fact of participating in activities in a senior citizen center is often a factor increasing the quality and meaning of life, and it is not related to being a grandparent.
Overall, the paper is well - written and I enjoyed reading it.
Reviewer 2 Report
The topic is interesting and the academic style in the manuscript is fine. The main problem regarding this manuscript is methodology and analysis section. These problems can be fixed so I will try to list my recommendations as follows:
1) First of all the authors should clearly state the research problems or they can write their hypotesis. On page 3, the authors mentioned the objective and they wrote that "In the present study was expected that parental warmth would be associated with the better scores in grandparenting functioning...." I think instead of this, they can formulate the hypotesis.
2) On page 3 the authors stated that "The research was 145 carried out in three large cities of Spain (Barcelona, Sevilla, and Valencia). But there is no information why and how these cities were selected? Please explain the rationale to choose these cities. In the following sentence it was stated that "The participants were recruited from senior citizen centers [31]." I think the given reference in number 31 is not related to this sentence. Please check this reference.
3) Another problem is about Results sections. This section should be revisited. It seems that the authors conducted correlation and regression analysis however, they did not give any information about the normality of the data. To be able to conduct these analysis normal distribution aof the data should be proven. Please give skewness and kurtosis values of the scales or provide Q-Q plots. Also before the regression analysis please check if there are multicollinearity problems among the scales.
Reviewer 3 Report
The paper is highly interesting, well structured and the analysis was performed very competently. However, I recommend that the authors consider the following issues that need to be addressed:
- please highlight the main challenges that have been encountered while conducting research, completing research tools, given that respondents were grandparents (aged from 60-89 years old); How were these challenges or difficulties managed at the individual level?
- specify very clearly the limits of their research;
- underline the practical implications of the research findings;
- provide more suggestions for future work in the field.
